# A Radiogenomics Ensemble to Predict EGFR and KRAS Mutations in NSCLC

**Silvia Moreno** [1,2,*], **Mario Bonfante** [1], **Eduardo Zurek** [2], **Dmitry Cherezov** [3], **Dmitry Goldgof** [3], **Lawrence Hall** [3] and **Matthew Schabath** [4]

1    Systems Engineering, Universidad Simon Bolivar, Barranquilla 080001, Colombia; mbonfante1@unisimonbolivar.edu.co
2    Systems Engineering, Universidad del Norte, Atlántico 080001, Colombia; ezurek@uninorte.edu.co
3    Computer Science and Engineering, University of South Florida, Tampa, FL 33620, USA; cherezov@mail.usf.edu (D.C.); goldgof@mail.usf.edu (D.G.); lohall@mail.usf.edu (L.H.)
4    Cancer Epidemiology, Moffit Cancer Center, Tampa, FL 33617, USA; Matthew.Schabath@moffitt.org
*    Correspondence: smoreno12@unisimonbolivar.edu.co; Tel.: +57-300-555-5132

**Abstract:** Lung cancer causes more deaths globally than any other type of cancer. To determine the best treatment, detecting EGFR and KRAS mutations is of interest. However, non-invasive ways to obtain this information are not available. Furthermore, many times there is a lack of big enough relevant public datasets, so the performance of single classifiers is not outstanding. In this paper, an ensemble approach is applied to increase the performance of EGFR and KRAS mutation prediction using a small dataset. A new voting scheme, Selective Class Average Voting (SCAV), is proposed and its performance is assessed both for machine learning models and CNNs. For the EGFR mutation, in the machine learning approach, there was an increase in the sensitivity from 0.66 to 0.75, and an increase in AUC from 0.68 to 0.70. With the deep learning approach, an AUC of 0.846 was obtained, and with SCAV, the accuracy of the model was increased from 0.80 to 0.857. For the KRAS mutation, both in the machine learning models (0.65 to 0.71 AUC) and the deep learning models (0.739 to 0.778 AUC), a significant increase in performance was found. The results obtained in this work show how to effectively learn from small image datasets to predict EGFR and KRAS mutations, and that using ensembles with SCAV increases the performance of machine learning classifiers and CNNs. The results provide confidence that as large datasets become available, tools to augment clinical capabilities can be fielded.

**Keywords:** radiogenomics; NSCLC; machine learning; EGFR; KRAS; ensembles; CNN

## 1. Introduction

Globally, lung cancer is the leading cause of cancer-related death in men and the second-leading cause in women. In 2018, an estimated 1.8 million lung cancer deaths occurred, with 1.2 million in men and over 576,000 in women, accounting for 1 in 5 cancer-related deaths worldwide [1]. Advances in precision medicine and genomic analyses have resulted in a paradigm shift whereby lung tumors are characterized and classified by biomarkers and genetic alterations (e.g., gene expression, mutations, amplifications, and rearrangements) that are critical to tumor growth and can be exploited with specific targeted agents or immune checkpoint inhibitors. However, there are many limitations of tissue-based biomarkers such as they can be subject to sampling bias due to the heterogeneous nature of tumors, the requirement of tumor specimens for biomarker testing, and the assays can take significant time and be expensive [2]. As such, high-throughput and minimally invasive methods that can improve current precision medicine is a critical need.

Liquid biopsy is a good alternative for a non-invasive way to detect EGFR and KRAS mutations. The use of surrogate sources of DNA, such as blood, serum, and plasma samples, which often contain circulating free tumor (cft) DNA or circulating tumor cells

(CTCs), is emerging as a new strategy for tumor genotyping [3]. However, this technique is pretty recent and still has some disadvantages. Different studies have also shown that the amount of cftDNA is correlated with disease stage, which may make it difficult to detect in early stages of cancer. Moreover, non-tumor cfDNA might derive from different processes including necrosis of normal tissues surrounding the tumor cells or lysis of leukocytes after blood collection, which may make mutation difficult to detect. Even when recent versions of liquid biopsy techniques have been approved for clinical use, the sensitivity (or True Positive Rate) of this test is still a weak point [3]. All these concerns provide space for the application of other non-invasive techniques that may be more effective in early stages of cancer and may provide higher sensitivity rates.

Quantitative image features, or radiomics, have the potential to complement and improve current precision medicine. Radiomic features are non-invasive, are extracted from standard-of-care images, and do not require timely and often expensive laboratory testing. Additionally, radiomic features are not subject to sampling bias since the entire tumor is analyzed and represents the phenotype of the entire tumor in 3D and not just the portion that was subjected to biomarker testing, and can be applied for all stages of cancer.

Radiogenomics is an emerging and important field because it utilizes radiomics to predict genetic mutations, gene expression, and protein expression [4]. In lung cancer, there has been particular interest in predicting EGFR and KRAS mutations [5]. Epidermal Growth Factor Receptor (EGFR) is a protein on the surface of cells that regulates signaling pathways to control cellular proliferation. According to Bethune et al. ([6], p. 1), "Overexpression of EGFR has been reported and implicated in the pathogenesis of many human malignancies, including Non-Small Cell Lung Cancer (NSCLC). Some studies have shown that EGFR expression in NSCLC is associated with reduced survival, frequent lymph node metastasis and poor chemosensitivity". Lung adenocarcinomas with mutated EGFR have a significant response to tyrosine kinase inhibitors [6], which makes the detection of this mutation significant in determining patient treatment. On the other hand, Kirsten Rat Sarcoma viral oncogene (KRAS) is also a well-known tumor driver. Mutations of this gene have proven to be a useful biomarker to predict resistance to EGFR-based therapeutics [7]. Furthermore, some studies have shown that KRAS can be targetable with promising results in phase III of NSCLC [8,9].

Other authors have previously tried to predict EGFR and KRAS mutations in Non-Small Cell Lung Cancer (NSCLC) from image features. In the work presented by Gevaert et al. [5], the authors attempted to predict these mutations from semantic image features provided by radiologists. A predictive model for the EGFR mutation was proposed that achieved an AUC of 0.89; however, conclusive results for the KRAS mutation were not obtained. Pinheiro et al. [10] also found a correlation between imaging features and mutation status for EGFR mutation (AUC of 0.745) but could not find the same for the KRAS mutation. On the other hand, Wang et al. [11] utilized semantic features to predict EGFR and KRAS mutation and found a significant correlation between EGFR and KRAS mutations and lesions with a low ground glass opacity (GGO). In particular, the authors found that L858R point mutations, exon 19 deletions, and KRAS mutations were more common in lesions with a lower GGO proportion ($p$ = 0.029, 0.027 and 0.018, respectively). Mei et al. [12] utilized texture features to predict mutations in EGFR at exon 19 and exon 21. The authors reported an AUC of 0.66 for predicting EGFR exon 21 mutation using a model that included sex, non-smoking status, and the Size Zone Non-Uniformity Normalized radiomic feature. Shiri et al. [13] created machine learning models from PET and CT image features to predict both EGFR and KRAS mutations. These authors obtained an AUC of 0.75 for both mutations in CT images by applying a combination of K-Best and a variance threshold feature selector with logistic regression. Incorporating PET kept AUC values around 0.74. Other authors that utilized features from PET and CT are Koyasu et al. [14]. These authors applied Random Forest and Gradient Tree Boosting to predict EGFR mutation, and obtained an AUC of 0.659 with the latter algorithm and seven types of imaging features. Liu et al. [15] utilized radiomics features and clinical data

to predict EGFR status and found an AUC of 0.647 with a model based on five radiomic features, which improved to 0.709 by combining radiomic features and clinical data. Deep learning has recently been applied in the diagnosis of different types of cancer [16], and other authors such as Wang et al. [17] have applied these techniques to mutation prediction. These authors utilized deep learning to the prediction of EGFR mutational status by training on 14,926 CT images and obtained an AUC of 0.81 on an independent validation cohort. Other recent studies have applied clinical nomograms to predict EGFR mutation status. In the work presented by Zhang et al. [18], the authors combined CT features and clinical risk factors and used them to build a prediction nomogram. They obtained a 0.74 AUC on the validation cohort.

On the other hand, previous studies have demonstrated that applying ensembles to predictive models tends to improve the performance of predictions [19]. An ensemble model is created by generating multiple models and combining them to produce an output classification. To combine the different models, a voting process is performed among them to determine the final result. There are different types of voting; for example, average voting, in which the average of the probabilities for each class of all the models is computed, and then a classification is performed based on the average probability. Another type of voting is maximum probability, in which for each case the base model with the higher pseudo-probability is selected, and the classification of the case is performed based on the pseudo-probabilities of this classifier alone.

In this paper, a novel voting scheme for ensembles of machine learning or deep learning models is proposed, and its effectiveness in predicting EGFR and KRAS mutations in CT images taken from the TCIA NSCLC Radiogenomics dataset [20] is shown to be state of the art. Two experiments were performed; first, prediction with radiomic features and machine learning models, and second, prediction through Convolutional Neural Networks (CNN). In both cases, first base models are tested and then an ensemble of the best models with a new voting scheme is applied to observe if there is an improvement of the prediction performance. Our approach shows that performance can be improved by this scheme and that good results are possible even with a small dataset where only a few cases present mutations. With more data becoming available in the future, it is expected that this type of approach will add to tools for clinicians.

## 2. Materials and Methods

For this study, a cohort of 99 patients from the TCIA were obtained [20,21], whose data included CT images with tumor segmentation on the CT image, genomic data (KRAS mutational status, and EGFR mutational status), and clinical data (age, sex, smoking status, pathological T stage, pathological N stage, pathological M stage, and histology type). Details of the cohort and corresponding data are published in a previous study [5]. Patients with unknown mutational status were eliminated from the analysis, which resulted in 83 patients for the analysis. The list of the exact cases that were used in the study can be found in the Supplementary Material (Table S7 Features Transpose EGFR, Table S8 Features Transpose KRAS). This type of data, with curation, is difficult to obtain. This set, while small, allows for comparisons. The summary of the study cohort is presented in Table 1. Table 2 summarizes the clinical features of the study cohort.

**Table 1.** Mutation status data summary.

| Variable | Values | Number of Cases (%) |
|---|---|---|
| EGFR Mutation Status | Mutant | 12 (14%) |
| | Wildtype | 71 (86%) |
| | Total | 83 (100%) |
| KRAS Mutation Status | Mutant | 20 (24%) |
| | Wildtype | 63 (76%) |
| | Total | 83 (100%) |

**Table 2.** Clinical features data summary.

| Variable. | Overall Dataset | EGFR Mutant | EGFR Wildtype | KRAS Mutant | KRAS Wildtype |
|---|---|---|---|---|---|
| Median Age (Range) | 69 (46–85) | 72 (55–85) | 69 (46–84) | 68 (50–81) | 69 (46–85) |
| **Gender** | | | | | |
| Male | 65 (78%) | 7 (8%) | 58 (70%) | 16 (19%) | 49 (59%) |
| Female | 18 (22%) | 5 (6%) | 13 (16%) | 4 (5%) | 14 (17%) |
| **Smoking Status** | | | | | |
| Current | 18 (22%) | 1 (1%) | 17 (21%) | 6 (6%) | 12 (16%) |
| Former | 56 (67%) | 8 (9%) | 48 (58%) | 14 (17%) | 42 (50%) |
| Non-smoker | 9 (11%) | 3 (4%) | 6 (7%) | 0 (0%) | 9 (11%) |
| **Pathological T Stage** | | | | | |
| Tis | 3 (4%) | 1 (1%) | 2 (3%) | 0 (0%) | 3 (4%) |
| T1a | 17 (21%) | 1 (1%) | 16 (20%) | 4 (5%) | 13 (16%) |
| T1b | 19 (23%) | 5 (6%) | 14 (17%) | 3 (3%) | 16 (20%) |
| T2a | 26 (31%) | 3 (3%) | 23 (28%) | 7 (8%) | 19 (23%) |
| T2b | 6 (7%) | 1 (1%) | 5 (6%) | 1 (1%) | 5 (6%) |
| T3 | 8 (9%) | 1 (1%) | 7 (8%) | 5 (6%) | 3 (3%) |
| T4 | 4 (5%) | 0 | 4 (5%) | 0 (0%) | 4 (5%) |
| **Pathological N Stage** | | | | | |
| N0 | 65 (78%) | 10 (12%) | 55 (66%) | 16 (20%) | 49 (58%) |
| N1 | 8 (10%) | 1 (1%) | 7 (9%) | 1 (1%) | 7 (9%) |
| N2 | 10 (12%) | 1 (1%) | 9 (11%) | 3 (3%) | 7 (9%) |
| **Pathological M Stage** | | | | | |
| M0 | 80 (96%) | 12 (14%) | 68 (82%) | 19 (23%) | 61 (73%) |
| M1b | 3 (4%) | 0 0% | 3 (4%) | 1 (1%) | 2 (3%) |
| **Histology** | | | | | |
| Adenocarcinoma | 66 (80%) | 12(14%) | 54 (66%) | 19 (23%) | 47 (57%) |
| Squamous cell carcinoma | 14 (17%) | 0 (0%) | 14 (17%) | 0 (0%) | 14 (17%) |
| NSCLC NOS | 3 (3%) | 0 (0%) | 3 (3%) | 1 (1%) | 2 (2%) |

For the EGFR mutation case, there is not a significant difference observed between the mutant and wildtype statuses in terms of age. In terms of gender, for the mutant status there seems to be a more balanced distribution between the genders, while the wildtype status seems to be significantly more common among men. In terms of smoking history, the EGFR mutant status seems to be found more often among former smokers, and non-smokers in second place, while the wildtype status seems to be more common among former and current smokers. There is no significant difference between the groups in terms of T cancer stage, although wildtype status seems to be more common for patients with stage T1a. Cases with stages N1 and N2 seem to more frequently present wildtype status, as well as

patients with M1b stage. In terms of histology type, none of the Squamous Cell Carcinoma patients present the EGFR mutation; this is only present in Adenocarcinoma cases.

For the KRAS mutation, there are no significant differences in terms of age and gender between the mutant and wildtype cases. In terms of smoking history, it can be observed that none of the non-smokers presented the KRAS mutation. For the pathological stage, it seems that most patients with stage N1 and N2 are wildtype cases. Moreover, as seen with the EGFR mutation, mutant status is only found in Adenocarcinoma. For more information about the distribution of the clinical variables in the Train and Test datasets, please refer to the Supplementary Material (Table S1. Clinical Variables Training Dataset, Table S2. Clinical Variables Test Dataset).

With this dataset two experiments were conducted; first, with traditional radiomic features and machine learning models, and second, with Convolutional Neural Networks (CNNs). Both experiments consisted of a base classifier performance assessment and then ensembles of several models were tested with three types of voting: average, maximum, and the method proposed here, Selective Class Average Voting (SCAV). SCAV is a voting technique that is particularly useful when dealing with an unbalanced dataset, where one class (majority class) is much more frequent than the other (minority class). In SCAV, first we count how many models predicted the minority class (in our case, the mutant status), and if this quantity is above a threshold value, the final outcome is the minority class. The pseudo-probability of this particular case is computed by averaging the scores of all the models where the final result was the minority class. If the value is below the chosen threshold, the final outcome is the majority class (in this case, the wildtype status), and the class pseudo probability is computed by finding the average of probabilities of all the models where the final result was the majority class. Once the probabilities are averaged according to the previous process, a threshold of 0.5 is applied to the final score to determine if the sample belongs to the minority (mutant) or to the majority (wildtype) class. To select the best thresholds for SCAV, that is the threshold for how many models must vote for the minority class, the performance of the ensemble on the Training set was assessed, and the thresholds that enabled a higher AUC on this data were selected and applied to the Test data. Figure 1 describes the algorithm used by SCAV.

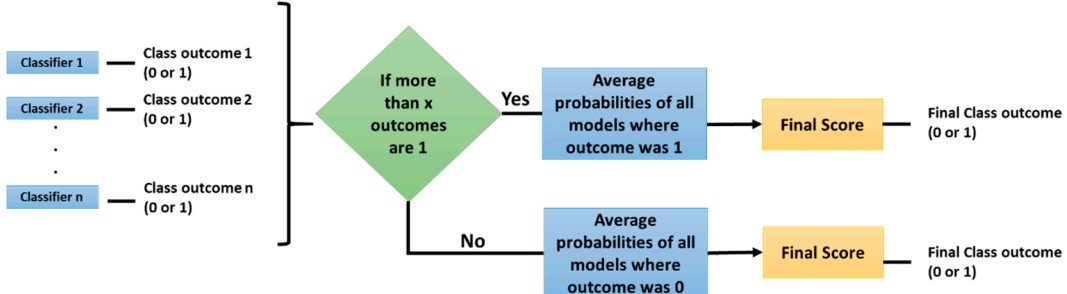

**Figure 1.** Selective Class Average Voting (SCAV) algorithm.

The use of ensembles increases the probability of obtaining better results, since we have several diverse models as inputs, and their errors tend to be out voted by the full set of classifiers. This enables better generalization error. However, there are some disadvantages to this approach; first that it consumes more time. Several models have to be trained before an ensemble can be attempted, and it requires more computing power and resources, since we have several classifiers running at the same time. This last item creates a limitation in how many total models can be used in the ensemble.

## 2.1. Experiment 1: Radiomic Features and Machine Learning Classifiers

Quantitative image features (N = 266) presented in [22] were extracted from the segmented 3D regions which included texture and non-texture features. These features were

computed using the segmented volumes publicly available in the NSCLC Radiogenomics dataset. Non-texture features include tumor size, tumor shape, and tumor location categories, and texture features include pixel histogram, run length, co-occurrence, Laws, and Wavelet features. To extract these features, Definiens Developer XD© (Munich, Germany) was used [23]. Definiens is based on the Cognition Network Technology that allows the development and execution of image analysis applications. Here, the Lung Tumor Analysis application was used. Most of the features were implemented within the Definiens platform, whereas some were computed with an implementation of the algorithms in C/C++ developed in a previous work by some of the authors of this paper [22].

For stage 1, the following experimental workflow was applied to predict mutation status from image features. First, the data was divided into Train and Test sets as part of a 10-fold cross validation. Second, on the Training set, feature selection was applied to select the image features with the most predictive power; third, the SMOTE algorithm [24] was applied to balance the number of examples in each class of the dataset; fourth, a classifier was trained with the previously selected features as inputs on the balanced Training data, and finally, the resulting model was applied to the Test set. Figure 2 summarizes the presented workflow.

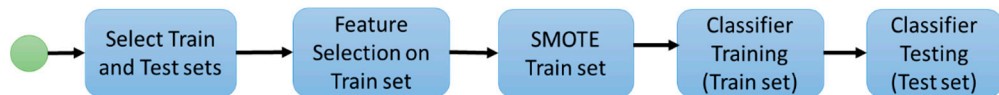

**Figure 2.** Experiment 1 workflow.

Feature selection was used to determine the image features with more predictive power. Sets of 5, 10, 15, and 20 features were tested. For feature selection, two methods were separately applied. The selection approaches were the Mann–Whitney test [25] and ReliefF [26]. Since this is a case of an unbalanced dataset (one class is much more abundant than the other), an optional application of the SMOTE algorithm [24] was performed to create synthetic samples of the minority class. The SMOTE algorithm was applied with the default settings. These settings make the dataset approximately balanced by class.

Finally, a classifier was trained with the selected features. Four machine learning classifiers were used: Random Forests [27], Support Vector Machines [28], Stochastic Gradient Boosting [29], and Neural Networks [30]. For every experiment, standard metrics were computed: including accuracy, sensitivity, and specificity (assuming the mutant status as the positive case), and Area Under the ROC Curve (AUC) [31]. The workflow was applied in a ten-fold cross-validation scheme, where iteratively nine folds were used to select features and train the classifier, and the left-out fold was used for testing the model.

The whole process was coded and executed in R 3.5.1 using the package FSelector [32] for the ReliefF feature selection, package DMwR [33] for the SMOTE algorithm, and package caret [34] to test the four different classifiers. These classifiers were executed with the default hyperparameters of the caret package. The implementation of the Mann–Whitney Feature Selector was coded in R.3.5.1 using the wilcox.test function to compute the p-value of every feature (every column of the dataset) and then features were sorted by this value in increasing order.

In the second stage of the experiment, an ensemble of base models was applied using a subset of the 32 learned models (obtained from 4 feature sets, 4 machine learning models, and 2 feature rankers). Sets of the top ranked 5, 10, and 20 existing models were tested, based on computational restrictions and the desire to have larger ensembles for typically better accuracy. To select which base models would be part of the ensemble, the average performance on the Training set was considered. The base models were sorted according to their average AUC on the Training set, and the top 5, 10, and 20 were selected.

### 2.2. Experiment 2: Convolutional Neural Networks

In the second experiment, CNNs were applied to the problem of predicting EGFR and KRAS mutations. CNNs are a type of deep neural network that have proven to be useful in detecting patterns on images [35]. From the same TCIA dataset, the CT images from the 83 patients that had both tumor segmentation and mutation information were selected and processed so a volume with only the tumor would be obtained. Then images of the Region of Interest (ROI) with a uniform size of 128 × 128 pixels per slice were extracted. From the whole volume, up to three slices per patient were selected to be part of the final dataset. The slice that had the largest tumor area was selected by manual visual inspection by the lead author of the segmented images. Then, we left one out in both directions of the *z*-axis and selected the two slices that where closest to the chosen slice up and down. The immediately consecutive slices were not used, assuming they were too similar to the central image. A slice without a clear piece of tumor in it was discarded. The dataset was then split into three: Training (65%), Validation (15%), and Test (20%) datasets. Since there was more than one image from each patient, we verified that images from the same patient were assigned to the same dataset.

In the first stage of the second experiment, several CNN models were applied to predict EGFR and KRAS mutations, varying conditions such as the CNN architecture, data augmentation, the optimizer, the learning rate, and the number of epochs of training. Since this is a very small dataset, small CNN architectures were tested. For the CNN experiments, we varied the CNN architecture (3 architectures), the optimizer (SGD and Adam), the Initial Learning rate (0.01, 0.005, and 0.0005), and the number of epochs (10, 20, and 30). Other numbers of epochs were also tested, based on the performance observed when the first three options were assessed. Furthermore, other architectures were tested (up to 10), but not with all the combinations. Figures 3–5 show the three best CNN architectures used in this experiment. The others can be found in Supplementary Material (Figure S1 CNN ARCHITECTURES).

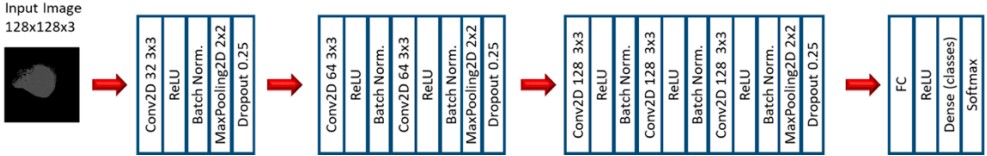

**Figure 3.** Architecture 1 of CNN base models.

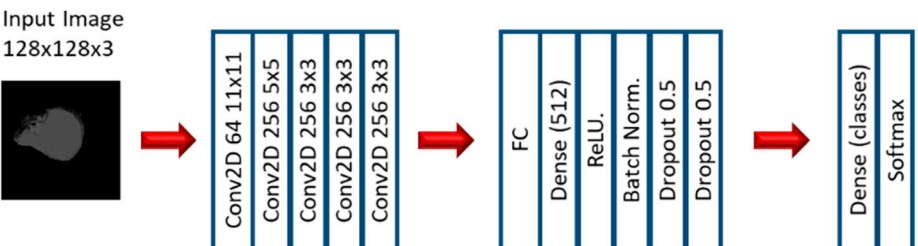

**Figure 4.** Architecture 4 of CNN base models.

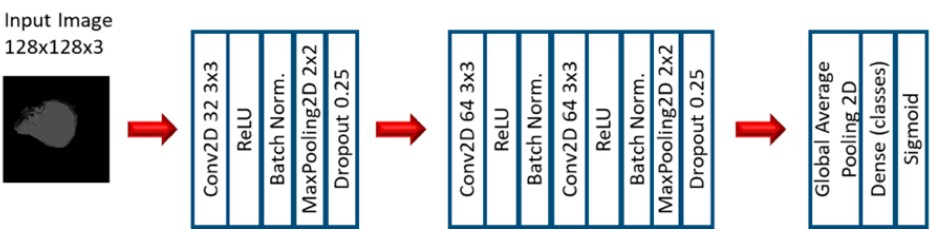

**Figure 5.** Architecture 6 of CNN base models.

More than 54 models were trained with different combinations of the before mentioned parameters. When enough good results were obtained using the base CNN models, a second stage of ensembles of CNN models was performed. Combinations of several models from the ones trained in the previous stage were tested. The models were ranked according to their performance on the Training set, and the best ones were selected for the ensembles. Different types of voting were applied: average, maximum, and SCAV.

The second experiment was coded in Python 3, and the library OpenCV was used for the image processing tasks. For the CNN generation, the library Keras with TensorFlow backend was utilized.

## 3. Results

### 3.1. Machine Learning Models: EGFR Mutation

The ten best results of the performance of the base classifiers for the EGFR mutation on the Test dataset are presented in Table 3, sorted by their AUC. The results on the Training set are included in the Supplementary Material (Table S3. EGFR Mutation Prediction Results Base Classifiers). The classifiers are Gradient Based Method (gbm), Random Forest (RF), Support Vector Machine (SVM), and Neural Network (nnet).

**Table 3.** EGFR mutation prediction results on Test dataset, base classifiers.

| Feature Selection | Classifier | SMOTE | Accuracy | Sensitivity | Specificity | AUC |
|---|---|---|---|---|---|---|
| MW (5 features) | nnet | No | 0.83 | 0.00 | 0.98 | 0.43 |
| ReliefF (15 features) | SVM | Yes | 0.76 | 0.66 | 0.78 | 0.68 |
| ReliefF (10 features) | RF | Yes | 0.76 | 0.41 | 0.82 | 0.67 |
| ReliefF (15 features) | nnet | Yes | 0.76 | 0.58 | 0.79 | 0.67 |
| ReliefF (5 features) | RF | Yes | 0.77 | 0.50 | 0.82 | 0.64 |
| ReliefF (20 features) | RF | Yes | 0.73 | 0.16 | 0.83 | 0.63 |
| ReliefF (20 features) | SVM | Yes | 0.68 | 0.66 | 0.69 | 0.63 |
| ReliefF (5 features) | nnet | Yes | 0.71 | 0.50 | 0.75 | 0.60 |
| ReliefF (5 features) | SVM | Yes | 0.79 | 0.25 | 0.89 | 0.59 |
| ReliefF (15 features) | RF | Yes | 0.72 | 0.25 | 0.80 | 0.57 |
| MW (5 features) | gbm | Yes | 0.68 | 0.16 | 0.78 | 0.53 |

The highest AUC for EGFR mutation prediction was 0.68 with an SVM classifier. For this mutation, much better results were obtained with ReliefF as feature selector.

Then, ensembles of different numbers of models with three different types of voting were tested. Table 4 presents the best results with ensembles. The best AUC was 0.70 with SCAV. This model also had the higher sensitivity (0.75). Moreover, in another model, an accuracy of 80% was obtained with a 0.68 AUC. It can be observed for the machine learning experiment that a higher accuracy, sensitivity, specificity, and AUC can be obtained by applying ensembles and SCAV. Different ensemble combinations can be used to favor certain metrics.

**Table 4.** EGFR mutation prediction best results on Test dataset, ensembles.

| Ensemble Combination | Classifiers | Accuracy | Sensitivity | Specificity | AUC |
|---|---|---|---|---|---|
| Ensemble SCAV thresh 3 (10 models) | gbm, SVM, nnet | 0.59 | 0.75 | 0.57 | 0.70 |
| Ensemble SCAV thresh 6 (10 models) | gbm, SVM, nnet | 0.80 | 0.33 | 0.89 | 0.68 |
| Ensemble Average (10 models) | gbm, SVM, nnet | 0.78 | 0.16 | 0.89 | 0.68 |
| Ensemble Average (5 models) | All | 0.78 | 0.16 | 0.89 | 0.67 |
| Ensemble Average (5 models) | RF, SVM, nnet | 0.79 | 0.33 | 0.87 | 0.66 |
| Ensemble Maximum (10 models) | gbm, SVM, nnet | 0.75 | 0.41 | 0.82 | 0.59 |

*3.2. Machine Learning Models: KRAS Mutation*

The results of the performance of the ten best base classifiers for the KRAS mutation on the Test dataset are presented in Table 5. The results on the Training set are included in the Supplementary Material (Table S4. KRAS Mutation Prediction Results, Base Classifiers). The best AUC is 0.65. For this mutation, similar results could be obtained with both feature selection methods, though ReliefF was still best.

**Table 5.** KRAS mutation prediction results on Test dataset, base classifiers.

| Feature Selection | Classifier | SMOTE | Accuracy | Sensitivity | Specificity | AUC |
|---|---|---|---|---|---|---|
| MW (10 features) | nnet | No | 0.72 | 0.10 | 0.93 | 0.44 |
| Relief (10 features) | nnet | No | 0.75 | 0.00 | 1.00 | 0.44 |
| ReliefF (5 features) | SVM | Yes | 0.70 | 0.35 | 0.81 | 0.65 |
| MW (15 features) | SVM | Yes | 0.64 | 0.40 | 0.72 | 0.64 |
| ReliefF (5 features) | gbm | Yes | 0.64 | 0.60 | 0.65 | 0.63 |
| ReliefF (20 features) | gbm | Yes | 0.63 | 0.50 | 0.67 | 0.63 |
| MW (10 features) | SVM | Yes | 0.64 | 0.45 | 0.70 | 0.63 |
| MW (20 features) | SVM | Yes | 0.71 | 0.40 | 0.81 | 0.63 |
| ReliefF (15 features) | SVM | Yes | 0.67 | 0.45 | 0.75 | 0.62 |
| MW (5 features) | SVM | Yes | 0.71 | 0.35 | 0.83 | 0.62 |
| MW (15 features) | gbm | Yes | 0.67 | 0.40 | 0.77 | 0.62 |
| ReliefF (15 features) | RF | Yes | 0.62 | 0.40 | 0.70 | 0.60 |

Then, ensembles of different numbers of models with three different types of voting were tested. Table 6 presents the best results with ensembles. The ensemble approach resulted in an improved AUC of 0.71 with a 72% accuracy using SCAV. Again, the models with best accuracy and best AUC were obtained with the proposed voting scheme. This was the best AUC that could be obtained for the KRAS mutation with the machine learning models.

**Table 6.** KRAS mutation prediction best results on Test dataset, ensembles.

| Ensemble Combination | Classifiers | Accuracy | Sensitivity | Specificity | AUC |
|---|---|---|---|---|---|
| Ensemble SCAV thresh 8 (10 models) | SVM, nnet | 0.72 | 0.20 | 0.89 | 0.71 |
| Ensemble SCAV thresh 6 (10 models) | SVM, nnet | 0.73 | 0.30 | 0.87 | 0.69 |
| Ensemble Average (5 models) | SVM | 0.70 | 0.35 | 0.81 | 0.67 |
| Ensemble Maximum (5 models) | SVM | 0.70 | 0.40 | 0.80 | 0.66 |
| Ensemble Average (10 models) | SVM, nnet | 0.66 | 0.35 | 0.76 | 0.65 |

### 3.3. Convolutional Neural Networks: EGFR Mutation

The best results of EGFR mutation prediction applying CNNs on the Test set are presented in Table 7. Please refer to the Supplementary Material (Table S5. EGFR Mutation Best Results, CNNs) for the results on the Train set. It can be observed that all the best results were obtained with SGD as optimizer; this suggests that SGD can be a good choice when dealing with small datasets with small CNN architectures. The best result was obtained with Architecture 4, which is presented on Figure 4. This model had an AUC of 0.846 and an accuracy of 0.800. This was the best AUC that could be obtained for the EGFR mutation.

**Table 7.** EGFR mutation best results on Test dataset, CNNs.

| Model | Optimizer | Learning Rate | Epochs | Accuracy | Sensitivity | Specificity | AUC |
|---|---|---|---|---|---|---|---|
| Arch. 4 | SGD | 0.0005 | 30 | 0.800 | 0.667 | 0.846 | 0.846 |
| Arch. 6 | SGD | 0.0005 | 30 | 0.771 | 0.222 | 0.961 | 0.752 |
| Arch. 1 | SGD | 0.01 | 8 | 0.400 | 1.000 | 0.192 | 0.688 |
| Arch. 6 | SGD | 0.01 | 10 | 0.657 | 0.666 | 0.654 | 0.675 |
| Arch. 3 | SGD | 0.01 | 7 | 0.543 | 0.778 | 0.461 | 0.671 |
| Arch. 4 | SGD | 0.01 | 10 | 0.543 | 0.778 | 0.461 | 0.628 |
| Arch. 1 | SGD | 0.01 | 30 | 0.514 | 0.778 | 0.423 | 0.623 |
| Arch. 2 | SGD | 0.01 | 30 | 0.542 | 0.667 | 0.538 | 0.571 |
| Arch. 4 | SGD | 0.01 | 20 | 0.600 | 0.444 | 0.654 | 0.559 |

After the base CNN models were obtained, an ensemble of the best CNN models was created. Table 8 presents the results of the best ensembles of CNN models. The best result in terms of AUC was 0.820 and an accuracy of 0.828, this result was obtained with a combination of the three best models and SCAV. An even better accuracy (0.857) was obtained with the combination of the five best models. This was the best accuracy for the EGFR mutation. Even if in this case there was not an increase in performance in terms of AUC, a better accuracy was obtained by applying SCAV.

**Table 8.** EGFR mutation best results on Test dataset, ensembles of CNNs.

| Model | Accuracy | Sensitivity | Specificity | AUC |
|---|---|---|---|---|
| Ensemble (3 models) SCAV thresh 3 | 0.828 | 0.667 | 0.885 | 0.820 |
| Ensemble (5 models) SCAV thresh 5 | 0.857 | 0.667 | 0.923 | 0.778 |
| Ensemble (3 models) Average | 0.486 | 0.778 | 0.385 | 0.743 |
| Ensemble (5 models) Average | 0.628 | 0.778 | 0.577 | 0.641 |
| Ensemble (3 models) Maximum | 0.371 | 0.778 | 0.231 | 0.624 |

*3.4. Convolutional Neural Networks: KRAS Mutation*

The best results of KRAS mutation prediction using CNNs on the Test set are presented in Table 9. The results on the Training set can be found in the Supplementary Material (Table S6. KRAS Mutation Best Results, CNNs). Analyzing the results for KRAS, we can see there is not a model that performs well according to all three metrics. The best result according to AUC is 0.739, however the sensitivity of this model is zero, so none of the mutant cases were detected. The model with the best accuracy has 72.2% and a sensitivity of 0.25, so it is a more balanced result, however the AUC is only 0.566.

**Table 9.** KRAS mutation best results on Test dataset, CNNs.

| Model | Optimizer | Learning Rate | Epochs | Accuracy | Sensitivity | Specificity | AUC |
|---|---|---|---|---|---|---|---|
| Arch. 1 | SGD | 0.01 | 60 | 0.667 | 0.000 | 1.000 | 0.739 |
| Arch. 6 | Adam | 0.005 | 10 | 0.333 | 1.000 | 0.000 | 0.607 |
| Arch. 6 | Adam | 0.001 | 10 | 0.667 | 0.000 | 1.000 | 0.593 |
| Arch. 1 | Adam | 0.005 | 15 | 0.722 | 0.250 | 0.958 | 0.566 |
| Arch. 1 | SGD | 0.01 | 90 | 0.667 | 0.000 | 1.000 | 0.555 |
| Arch. 1 | SGD | 0.01 | 10 | 0.555 | 0.667 | 0.500 | 0.531 |

In order to improve the results, an ensemble of the best CNN models was created. Table 10 shows the best results with ensembles of CNNs. The best AUC that could be obtained in this stage was 0.778, which was obtained with an ensemble of the three best models and average voting. This was the best AUC that could be obtained for the KRAS mutation. This was the only stage where the best results in terms of AUC were not obtained with SCAV; however, an equal accuracy could be obtained applying SCAV with the best three models.

**Table 10.** KRAS mutation best results on Test dataset, ensembles of CNNs.

| Model | Accuracy | Sensitivity | Specificity | AUC |
|---|---|---|---|---|
| Ensemble (3 models) Average | 0.722 | 0.250 | 0.958 | 0.778 |
| Ensemble (3 models) SCAV thresh 2 | 0.722 | 0.250 | 0.958 | 0.722 |
| Ensemble (4 models) SCAV thresh 3 | 0.722 | 0.250 | 0.958 | 0.642 |
| Ensemble (7 models) SCAV thresh 4 | 0.694 | 0.416 | 0.833 | 0.618 |
| Ensemble (7 models) SCAV thresh 5 | 0.694 | 0.083 | 1.000 | 0.604 |

## 4. Discussion

*4.1. EGFR Mutation*

Several observations can be made from these results. Our first observation is that for the machine learning models, the use of SMOTE greatly improves the performance of the models when dealing with unbalanced datasets. Without the SMOTE algorithm, the sensitivity was zero, but while using it several of the mutant cases were properly detected.

For the EGFR mutation, good results could be obtained with the machine learning approach; however, despite the small training dataset, better results could be obtained with CNNs. In both cases, the base models could be improved by using ensembles. For the machine learning models, the best base performance was with ReliefF as feature selector, 15 features, and SVM as the classifier.

For the machine learning approach, better results in terms of accuracy, sensitivity, and AUC could be obtained with different combinations of ensembles and SCAV. The best features in the sense that they were more commonly selected in the best machine learning models are: 3D Wavelet features, 3D Laws features, GLSZM Grey level variance, 90th percentile, GLSZM Small zone low grey level emphasis, Flatness, Asymmetry, Orientation, and Surface to volume ratio.

In the tests applying CNNs, the best performance of the base models was with Architecture 4 and SGD as the optimizer. After applying ensembles, we got an improvement in accuracy. This result was obtained using our proposed voting scheme, SCAV. In general, we obtained better results working with CNNs than with the machine learning models. We also observed that the ensembles with SCAV outperformed the ones with average and maximum voting. This suggests that the proposed scheme can obtain better performance when applying ensembles, even if the performance of the base models is not optimal.

Our results, where features are extracted automatically, while slightly less than the AUC of 0.89 obtained by Gevaert et al. [5] with the same dataset, did not require medical experts to produce semantic features. However, our AUC is superior to the ones obtained by other previous works with an automated approach. Other metrics such as accuracy were not reported and cannot be compared.

### 4.2. KRAS Mutation

For the base classifiers of the machine learning approach, good results could be obtained both with the ReliefF feature selector and the Mann–Whitney test. The best AUC was obtained with a model of ReliefF's best five features and SVM as the classifier. The best features in the sense that they were more commonly selected in this model were 3D Laws features, 3D Wavelet features, average GLN Grey level non-uniformity, and GLSZM Grey level non-uniformity. With the ensemble approach, an important increase in the performance was obtained by applying SCAV as the voting scheme. The best results with ensembles and machine learning models were always obtained with SCAV as the voting scheme. This shows that ensembles can significantly improve the performance of classifiers.

For the CNN models, there were no models that achieved high scores in all the three metrics (accuracy, sensitivity, and AUC). The best AUC was obtained with Architecture 1 and SGD as the optimizer, and the best accuracy was with Architecture 1 and Adam as the optimizer. After applying ensembles of CNN models, the performance improved in AUC and the accuracy was maintained. In this case, the best result was obtained with average voting; however, the second best result was obtained with SCAV. It can be observed that in terms of AUC, better results could be obtained with CNNs over machine learning models. In both cases, an improvement was observed with the ensemble approach.

If we compare these results with the ones obtained by Gevaert et al. [5] with the same dataset, the authors could not find a conclusive model for KRAS mutation with semantic features (AUC of 0.55). We did find a good predictive model for KRAS mutation on the same dataset with an AUC of 0.778 and using a deep learning approach that does not need human input to generate features.

### 4.3. Limitations

A limitation of this study is the small dataset. Furthermore, even if separate training, validation, and test datasets are used, it would be more conclusive if the trained model could be tested on a dataset from a different source, which would prove the generalization of the model. Both these limitations will be addressed as future work, when more data that fits the requirements of the study is available. Finally, for the machine learning approach,

because we chose features on each fold of a cross validation, the very best set is the one that occurred most often and could differ when more data is available.

## 5. Conclusions

In this study, we analyzed the effectiveness of using ensembles in the prediction of EGFR and KRAS mutations using a small dataset; in particular, we assessed the performance of a novel voting scheme SCAV. We tested this scheme with both ensembles of machine learning models and ensembles of CNNs and a significant improvement from the base classifiers was observed.

For the EGFR mutation, the performance of our model was similar to that obtained by Gevaert et al. with the same dataset, and our model did not require semantic features manually specified by a radiologist. Further, our best model obtained a higher AUC than the ones presented by the most recent works that used deep learning [17] and nomograms [18].

For the KRAS mutation, the results are much better than the ones obtained in [5], where a conclusive model for KRAS mutation could not be found for this same dataset. Moreover, this is probably the best result for the KRAS mutation prediction that can be found in the literature, since most works only focus on the EGFR mutation.

In general, for both mutations, better results were be obtained by applying ensembles with SCAV as the voting method, rather than average and maximum voting; however, a more rigorous method to determine the best threshold is still necessary. This work indicates that applying ensembles and SCAV for voting may lead to a significant increase in the performance of the base models, both for machine learning and deep learning models, which offers a good strategy to handle small datasets when no more data is available. Furthermore, higher sensitivity was obtained when applying the SMOTE algorithm for the machine learning models, which is an effective strategy to handle unbalanced classification datasets.

This work showed novel ways to use ensembles of CNNs and non-neural classifiers on small data to achieve state-of-the-art results. Our proposed approach, which is to use ensembles with SCAV, shows in this study that the performance of classifiers can be improved, even when the base models do not perform that well, and this is an important contribution from this paper. Since larger datasets will enable better models, we firmly believe that if our approach is applied with more data it will yield outstanding performance and may generate models that can be used in clinical practice. This indicates a promising future for detecting these mutations in a non-invasive way.

**Supplementary Materials:** The following are available online at https://www.mdpi.com/article/10.3390/tomography7020014/s1, Table S1. Clinical Variables Training Dataset, Table S2. Clinical Variables Test Dataset, Table S3. EGFR Mutation Prediction Results Base Classifiers, Table S4. KRAS Mutation Prediction Results, Base Classifiers, Table S5. EGFR Mutation Best Results, CNNs, Table S6. KRAS Mutation Best Results, CNNs, Table S7 Features Transpose EGFR, Table S8 Features Transpose KRAS, Figure S1 CNN ARCHITECTURES.

**Author Contributions:** Conceptualization, all authors; methodology, D.G., L.H., and M.S.; software, S.M., M.B., and D.C., validation, S.M. and E.Z.; formal analysis, S.M., D.G., L.H., and M.S.; investigation, S.M.; resources, D.G., L.H., and M.S.; data curation, S.M., M.B., and D.C.; writing—original draft preparation, S.M. and E.Z.; writing—review and editing, D.G., L.H., and M.S.; visualization, S.M. and M.B.; supervision, E.Z., D.G., and L.H.; project administration, S.M. and E.Z. All authors have read and agreed to the published version of the manuscript.

**Funding:** This research is partially supported by the National Institute of Health under grants (NIH U01 CA143062), (NIH U24 CA180927), and (NIH U01 CA200464).

**Institutional Review Board Statement:** Not applicable.

**Informed Consent Statement:** Not applicable.

**Data Availability Statement:** The dataset for this study is a subset from the one found at https://wiki.cancerimagingarchive.net/display/Public/NSCLC+Radiogenomics (accessed on 20 August 2018). Supplementary material indicates the exact data used from the set.

**Acknowledgments:** This research could not have been performed without the aid of the Fulbright Visiting Researcher Scholarship that allowed the collaboration between the University of South Florida and Universidad del Norte in Colombia. Doctoral studies of the first author are also supported by Universidad del Norte and Universidad Simon Bolivar.

**Conflicts of Interest:** The authors declare no conflict of interest. The funders had no role in the design of the study; in the collection, analyses, or interpretation of data; in the writing of the manuscript, or in the decision to publish the results.

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
