# Peer review of "A Radiogenomics Ensemble to Predict EGFR and KRAS Mutations in NSCLC"

_tomography, doi:10.3390/tomography7020014_

Round 1

Reviewer 1 Report

This paper highlights a new method (SCAV) as a voting scheme for ensembles, which seems useful for processing small amounts of data, often encountered in monocentric studies with small sample sizes and unbalanced classification datasets. The authors applied this model in lung cancer by using CT images from a radiogenomic dataset to evaluate its effectiveness (therefore the performance of predictions) in predicting EGFR and KRAS mutations. They wisely compared it to two classical ways of combining predictions (average and maximum voting) and showed that the diagnostic performance obtained with the proposed scheme is better than two others for machine learning and deep learning models. The article well reflects the current knowledge about voting ensembles and is also consistent with current knowledge in the field of lung cancer. In addition, it showed that this predictive model could be applied for predicting recurrent genomic alterations in lung cancer from CT images.

Minor revisions :

  1. Methods are clearly identified with a well depicted voting scheme and results are sufficiently developed, according to the different categories tested. However, the conclusions should be much more concised and the results should no longer be discussed in the latter. Indeed, repetitions are observed. Please modify.
  2. The only limitation mentioned is a more rigorous method needed to determine the best threshold. One unmentioned limitation is probably that the cohort does not contain CT-scan "negative" patients (e.g., CT-scan features that are suspicious but without proven cancer). Therefore, the performance obtained were probably overestimated since there is a selection bias. Indeed, the prediction model was not evaluated on negative or doubtful patients on CT-scan (false positive results could have been obtained by the predictive model). Other more general limitations related to the use of voting schemes should also be mentioned.
  3. The authors conclude that applying the correct methodology allows to build good predictive models even on small samples. Indeed, it will be necessary to evaluate these methods on larger samples and compare the use of the SMOTE algorithm for unbalanced classification datasets vs. balanced classification datasets.

Author Response

Thank you for giving us the opportunity to submit a revised draft of the manuscript “A Radiogenomics Ensemble to Predict EGFR and KRAS Mutations in NSCLC” for publication in “Tomography". We appreciate the time and effort that you dedicated to providing feedback on our manuscript and are grateful for the insightful comments on and valuable pointers for improvements to our paper.

Reviewers' Comments to the Authors:

“This paper highlights a new method (SCAV) as a voting scheme for ensembles, which seems useful for processing small amounts of data, often encountered in monocentric studies with small sample sizes and unbalanced classification datasets. The authors applied this model in lung cancer by using CT images from a radiogenomic dataset to evaluate its effectiveness (therefore the performance of predictions) in predicting EGFR and KRAS mutations. They wisely compared it to two classical ways of combining predictions (average and maximum voting) and showed that the diagnostic performance obtained with the proposed scheme is better than two others for machine learning and deep learning models. The article well reflects the current knowledge about voting ensembles and is also consistent with current knowledge in the field of lung cancer. In addition, it showed that this predictive model could be applied for predicting recurrent genomic alterations in lung cancer from CT images.”

Response: Thanks for your very accurate and positive summary.

Minor revisions :

1. Methods are clearly identified with a well depicted voting scheme and results are sufficiently developed, according to the different categories tested. However, the conclusions should be much more concised and the results should no longer be discussed in the latter. Indeed, repetitions are observed. Please modify.

Response: We have modified the conclusions section to be more concrete and avoid repetitions.

2. The only limitation mentioned is a more rigorous method needed to determine the best threshold. One unmentioned limitation is probably that the cohort does not contain CT-scan "negative" patients (e.g., CT-scan features that are suspicious but without proven cancer). Therefore, the performance obtained were probably overestimated since there is a selection bias. Indeed, the prediction model was not evaluated on negative or doubtful patients on CT-scan (false positive results could have been obtained by the predictive model). Other more general limitations related to the use of voting schemes should also be mentioned.

Response: Thanks for your insightful comments.  On the limitations of voting schemes, a paragraph about them has been included at the beginning of page 5.

 We do not believe that the absence of non-cancer patients is a limitation. The EGFR and KRAS mutation only become of interest when lung cancer is present, so it makes sense that only lung cancer images are included. Before testing for the mutations there should be another classifier that determines if a nodule is malignant or not, but that is out of the scope of this work.

3. The authors conclude that applying the correct methodology allows to build good predictive models even on small samples. Indeed, it will be necessary to evaluate these methods on larger samples and compare the use of the SMOTE algorithm for unbalanced classification datasets vs. balanced classification datasets.

 Response: Availability of public datasets that have CT data, tumor segmentations and mutation status is indeed an issue with this study. However we firmly believe that the positive results obtained with our small dataset are an indication that applying these methods to a bigger dataset will yield outstanding results. We are constantly looking for new datasets with all the requirements to validate our work, and will probably do so if the opportunity arises.

Reviewer 2 Report

 A Radiogenomics Ensemble to Predict EGFR and KRAS Mutations in NSCLC

The authors described machine learning in predicting EGFR and KRAS mutation in CT images. This is interesting paper. Thus, it worth to be published.

Minor

We don’t know what the machine learning used to predict the mutation. The authors had better provide CT images whose EGFR and KRAS mutation are positive or negative.  

Author Response

Thank you for giving us the opportunity to submit a revised draft of the manuscript “A Radiogenomics Ensemble to Predict EGFR and KRAS Mutations in NSCLC” for publication in “Tomography". We appreciate the time and effort that you dedicated to providing feedback on our manuscript and are grateful for the insightful comments on and valuable pointers for improvements to our paper.

Reviewers' Comments to the Authors:

The authors described machine learning in predicting EGFR and KRAS mutation in CT images. This is interesting paper. Thus, it worth to be published.

Response: Thanks for your positive comments.

Minor

We don’t know what the machine learning used to predict the mutation.

 Response: Since, we have small data and do a cross validation selecting features per fold, the set of features varies.  We have been more explicit in labeling best features, now. In the Discussion Section (Page 11, line 13) we note that for the EGFR mutation, the features that were more commonly selected in the best machine learning models were 3D Wavelet features, 3D Laws features, GLSZM Grey level variance, 90th percentile, GLSZM Small zone low grey level emphasis, Flatness, Asymmetry, Orientation and Surface to volume ratio. For the KRAS mutation (Page 11, line 35), we mention that the features that were more commonly selected in this model were 3D Laws features, 3D Wavelet features, average GLN Grey level non-uniformity and GLSZM Grey level non-uniformity.

The authors had better provide CT images whose EGFR and KRAS mutation are positive or negative.

Response: The CT images and the Clinical Data with Mutation status are publicly available at the TCIA website

https://wiki.cancerimagingarchive.net/display/Public/NSCLC+Radiogenomics.

To verify the subset that was used in this study please refer to the supplementary files: Features_Transpose_EGFR.xlsx and  Features_Transpose_KRAS.xlsx.

We added: “Supplementary material indicates the exact data used from the set” to the data availability section.

Reviewer 3 Report

The article presented is really interesting. It used image datasets to predict KRAS and EGFR mutations in NSCLC. It is also a well written manuscript. I am not sure of the future applicability of the work, since you need to evaluate a much larger sample size to further confirm your findings.

I suggest to improve figure 1, it is a bit small and hard to see the letters on it.

I am not sure if the discussion allows subtopics, please check with editor. The conclusions are too long, maybe part of the conclusions could be placed in the discussion. Please, make a more concise conclusion with the main findings.

Revise grammar and typos.

Author Response

Thank you for giving us the opportunity to submit a revised draft of the manuscript “A Radiogenomics Ensemble to Predict EGFR and KRAS Mutations in NSCLC” for publication in “Tomography". We appreciate the time and effort that you dedicated to providing feedback on our manuscript and are grateful for the insightful comments on and valuable pointers for improvements to our paper.

Reviewers' Comments to the Authors:

The article presented is really interesting. It used image datasets to predict KRAS and EGFR mutations in NSCLC. It is also a well written manuscript. I am not sure of the future applicability of the work, since you need to evaluate a much larger sample size to further confirm your findings.

Response: Thanks for your positive comments. We are indeed looking forward to validating our work when more data is available.

I suggest to improve figure 1, it is a bit small and hard to see the letters on it.

Response: We have updated Figure 1 to make it easier to read.

I am not sure if the discussion allows subtopics, please check with editor.

Response: We checked the instructions for Authors and it does not say anything against subsections. We also asked the Assistant Editor about the topic and she said that after the reviewing process the manuscript will be checked by the academic editor for format issues.

The conclusions are too long, maybe part of the conclusions could be placed in the discussion. Please, make a more concise conclusion with the main findings. Revise grammar and typos.

Response: We have modified the conclusions section to be more concrete and avoid repetitions.

Reviewer 4 Report

Moreno et al describe a radiogenomics ensemble to predict EGFR and KRAS mutations in non-small cell lung cancer. The manuscript is generally well-written, the analysis valid and the subject very interesting and clinically relevant, because mutation status is currently used for treatment stratification of newly diagnosed NSCLC patients.
I suggest to add a short paragraph about potential limitations of the study, for example the small number of cases and the lack of separate training and validation/test sets.
Do the authors maybe have access to data from their own institutions (Simon Bolivar, del Norte, USF/Moffit), which they could possibly use for validation of the results?

Author Response

Thank you for giving us the opportunity to submit a revised draft of the manuscript “A Radiogenomics Ensemble to Predict EGFR and KRAS Mutations in NSCLC” for publication in “Tomography". We appreciate the time and effort that you dedicated to providing feedback on our manuscript and are grateful for the insightful comments on and valuable pointers for improvements to our paper.

Reviewers' Comments to the Authors:

Moreno et al describe a radiogenomics ensemble to predict EGFR and KRAS mutations in non-small cell lung cancer. The manuscript is generally well-written, the analysis valid and the subject very interesting and clinically relevant, because mutation status is currently used for treatment stratification of newly diagnosed NSCLC patients.

Response: Thanks for your positive comments.

I suggest to add a short paragraph about potential limitations of the study, for example the small number of cases and the lack of separate training and validation/test sets.

Response: A paragraph about the limitations of the study has been added to the Discussion section (Page 11).

Do the authors maybe have access to data from their own institutions (Simon Bolivar, del Norte, USF/Moffit), which they could possibly use for validation of the results?

Response: We currently don’t have an additional dataset that fits all the requirements of the study and does not have restrictions on its use. However, we are constantly looking for additional datasets to validate our models, so this will be part of future work.